# The Effect of Loneliness on Work Engagement among Chinese Seafarers: Mediating Effects of Executive Functions

**DOI:** 10.3390/bs14100880

**Published:** 2024-10-01

**Authors:** Xinjie Qi, Daoke Li, Rong Lian

**Affiliations:** 1School of Psychology, Fujian Normal University, Fuzhou 350007, China; 2012033@fjcpc.edu.cn; 2School of Navigation, Fujian Chuanzheng Communications College, Fuzhou 350108, China; 2003013@fjcpc.edu.cn

**Keywords:** Chinese seafarers, loneliness, work engagement, executive function

## Abstract

In order to explore the effect of loneliness on work engagement and the role of executive function in Chinese seafarers, two studies were carried out. Study 1 conducted a questionnaire survey of 1231 active Chinese seafarers to examine the relationship between seafarers’ loneliness, work engagement, and executive functions. Study 2 involved 177 seafarers as participants and created scenarios of seafarer work loneliness and non-loneliness using a recall paradigm, measuring components of executive function such as inhibitory control, working memory, and cognitive flexibility, as well as willingness to engage in work. The findings indicate that seafarers’ sense of loneliness can significantly negatively predict their work engagement, with inhibition control, working memory, and cognitive flexibility playing mediating roles. This provides new references and insights for alleviating the negative impact of loneliness on seafarers and enhancing their work engagement through cognitive approaches.

## 1. Introduction

Safety is the foundation of the maritime industry, with human factors contributing to about 80% of navigation incidents [1]. Enhancing work engagement has been found to effectively reduce the occurrence of human-related safety accidents [2]. Work engagement consists of three dimensions: vigor, dedication, and absorption, reflecting an active state in an individual’s physical, cognitive, and emotional aspects at work [3]. This active state has been positively correlated with employee health and work performance [4]. Nevertheless, previous research has predominantly focused on descriptions of seafarers’ mental health [5]. Only one research has attempted to construct a model of the relationship between seafarers’ work and mental health through questionnaire data [2], but it lacks causal inference. Current research on work engagement is limited mostly to special professions such as police and prison officers [6,7], which offers only minimal reference value for broader contexts. Therefore, exploring the factors influencing seafarers’ work engagement and its internal mechanics is crucial for the development of maritime enterprises and holds significant practical value.

## 2. Literature Review and Theoretical Foundation

### 2.1. The Relationship between Seafarers’ Loneliness and Work Engagement

Social isolation is one of the key work characteristics of seafarers that separates them from normal social life. Loneliness and social isolation represent the objective and subjective deficits in social relational networks, respectively [8]. Loneliness refers to the negative emotional state that arises when an individual’s social needs are not met [9]. Since the COVID-19 pandemic, Chinese seafarers on international voyages have experienced high levels of loneliness, which has had numerous adverse effects on their sleep, physical health, and well-being [10]. Although existing research indicates that loneliness in the general workplace can reduce an individual’s level of work engagement [11], these studies focus on the general working population and their conclusions cannot be directly generalized to seafarers. Therefore, it is necessary to explore the influence of loneliness on work engagement among seafarers and the underlying mechanisms.

According to the Job Demands–Resources theory [12], when an individual’s resources are insufficient to meet job demands, resource depletion occurs, leading to reduced work engagement [11,13]. From the perspective of Affective Events Theory [14], the unique maritime working environment is likely to induce feelings of loneliness, impairing the emotional motivation component of seafarers’ work engagement. Additionally, the continuous experience of loneliness may compel seafarers to reduce their work engagement as a means to balance the gap between actual work perceptions and expected work aspirations, leading to dissatisfaction [15]. Finally, lonely seafarers might increasingly concentrate on strategies to satisfy their interpersonal needs as a way to counteract their dwindling sense of belonging [16]. This redirection of focus can monopolize their limited cognitive resources, ultimately leading to a reduction in their work engagement.

Although loneliness may increase an individual’s motivation to re-establish connections with others, according to Social Reconnection Theory [17], individuals will only increase their investment in relationships when they believe there is a possibility of forming interpersonal connections [18]. Seafarers work with a fixed set of colleagues and, thus, have a lower possibility of re-establishing interpersonal relationships. As a result, they may not invest cognitive resources in rebuilding these relationships but might instead turn to other avenues that can compensate for and meet their self-needs. Self-Determination Theory posits that individuals will compensate for their needs through other means [19,20,21]. Furthermore, from an evolutionary standpoint, environmental factors can reshape brain function [22]. Wang and colleagues, using fMRI technology, demonstrated that the auditory, visual, executive control, and vestibular networks in seafarers’ brains differ significantly from those of non-seafarers [23]. This suggests that seafarers have some tolerance to their work environment. Considering the aforementioned reasoning, this study does not hypothesize a specific direction of effect of loneliness on seafarers’ work engagement but merely proposes the following hypothesis:

**Hypothesis 1.** *Loneliness significantly influences the work engagement of seafarers*. 

### 2.2. The Mediating Role of Executive Functioning

Working in isolation, confined, and extreme (ICE) conditions, seafarers confront unique stresses where individual executive functions become especially critical [24]. Executive functions are top-down mental processes that control and manage information during cognitive activities, typically including inhibitory control, working memory, and cognitive flexibility [25]. Inhibitory control allows an individual to successfully achieve goals by suppressing internal or external distractions. This skill aids seafarers in consciously suppressing interference or reaction tendencies during goal-directed activities, which is closely linked to their capacity to handle emergencies [26]. Cognitive flexibility is the ability to adapt thinking processes according to situational demands and personal goals [27]. Research has shown it is associated with a range of positive cognitive capabilities such as concentrating, inhibiting irrelevant stimuli, and better controlling thoughts without falling into maladaptive, rigid mental routines [28]. Cognitive flexibility has been proven to significantly mediate the relationship between loneliness and psychological adaptation [29]. Working memory, responsible for temporarily storing and processing information [30], has been found to be a strong predictor for problem-solving and dealing with complex tasks among seafarers. Sims and his colleagues, using the Stroop and Wisconsin Card Sorting Task, found that actual social support could significantly predict inhibitory control and cognitive flexibility among middle-aged African Americans [31]. Dickinson and his colleagues, utilizing an Alzheimer’s cognitive questionnaire, found that declines in elderly individuals’ verbal working memory were associated with decreased social support [32]. A meta-analysis also indicated that loneliness could impair individual executive functions [33]. Moreover, according to the Limited Cognitive Resource Theory [34], seafarers coping with loneliness might deplete limited cognitive resources, weakening their executive functioning. Therefore, we proposed the following hypothesis:

**Hypothesis 2.** *Loneliness significantly negatively predicts the executive functions of seafarers*.

Grounded in the Job Demands–Resources Theory [35], executive functions can be conceptualized as a set of personal resources that facilitate coping with occupational demands, thereby potentially enhancing seafarers’ work engagement. Previous research has indicated an overall positive association between global executive functions, as measured by questionnaires, and work engagement across a general employed population [36]. However, this investigation did not dissect the specific contributions of executive function’s distinct subcomponents. Executive functions are generally posited to have a salutary effect on emotion regulation. Previous empirical findings have substantiated a positive linkage between executive functioning and emotion regulation capabilities [37,38]. On one hand, positive emotions can directly enhance the emotional dimension of work engagement. On the other hand, proficient emotion regulation is a key component of affective motivation, which drives work engagement [39,40]. Therefore, it can be inferred that executive functions may enhance work engagement by supporting effective emotion regulation.

Nevertheless, previous research has presented a lack of consensus concerning the specific roles played by the tripartite subcomponents of executive functions—inhibitory control, working memory, and cognitive flexibility—in the context of emotion regulation. For instance, Liu and her colleagues found inhibitory control and working memory to be predictive of better emotion regulation [41]. Alternatively, Guo and his colleagues found that individuals with higher levels of inhibitory control experience an advantage in recalling information when faced with emotional distractions [42]. Gao contended that cognitive flexibility supports the utilization and transition between emotion regulation strategies [43], while Michelle and Buchanan observed that only working memory was related to the regulation of negative emotions, with inhibitory control and cognitive flexibility not displaying a significant modulate effect on negative emotions [44].

These varied findings alert us to the possibility that the effect of executive function subcomponents on loneliness and work engagement among seafarers may be inconsistent. Therefore, we posit: 

**Hypothesis 3.** *That executive functions positively predict seafarers’ work engagement*. 

However, it does not make specific predictions about the impact of each individual subcomponent of executive function. Moreover, the outcomes of executive function as measured by scales are primarily indicative of an individual’s level of psychological reflective analysis, representing the extent to which individuals pursue goal attainment and rational decision making within daily life contexts. These measures may not fully capture cognitive efficiency or optimal behavioral performance in specific situational contexts [45]. Therefore, a combination of questionnaire assessments and behavioral experiments is necessitated to comprehensively measure the executive functions of seafarers. In Study 1, we will conduct an initial exploration of the relationships between seafarer loneliness, executive functions, and work engagement using a questionnaire survey method. In Study 2, we will manipulate seafarer loneliness through an experimental approach to further investigate the causal relationships between seafarer loneliness, executive functions, and work engagement. Additionally, we will examine the distinct roles of the subcomponents of executive functions in these relationships, providing a detailed analysis of each aspect.

## 3. Study 1

### 3.1. Participants

A total of 1356 seafarers were drawn from five crew service companies and three shipping enterprises, all of whom had served aboard a ship within the past year. Surveys were distributed and collected through the Questionnaire Star platform from March 10th to 18th, 2023. A total of 1258 questionnaires were collected, and after screening, 1231 valid questionnaires were identified for analysis. Because seafarers work in offshore or international waters, we first obtained verbal consent from the shipping company online. After the crew members completed their voyage and returned to land, we then collected their written informed consent.

### 3.2. Material and Methods

#### 3.2.1. Measurement of Work Engagement

The Chinese version of the Work Engagement Scale [46] was utilized for this measurement. This scale comprises 17 items across three dimensions: vigor, dedication, and absorption. A 5-point rating scale is used (1 = completely disagree, 5 = completely agree), with higher scores indicating greater work engagement. In this study, the Cronbach’s α reliability coefficient of this scale was 0.95.

#### 3.2.2. Measurement of Loneliness

The Chinese version of the Loneliness Scale (3rd edition) [47] was utilized for this measurement. This scale, which is composed of 20 items, utilizes a 4-point scoring system ranging from 1 (never) to 4 (always). Higher scores reflect higher levels of loneliness. In this research, the Cronbach’s α reliability coefficient for this scale was 0.82.

#### 3.2.3. Measurement of Executive Function

The Adult Executive Functioning Self-Assessment Scale [48] was employed. It is divided into two subscales: working memory and inhibitory control, with a total of 14 items rated on a 5-point scale (1 = completely agree, 5 = completely disagree). Higher scores indicate better executive functioning. In this study, the Cronbach’s α reliability coefficient for this scale was 0.92.

### 3.3. Results

Data were managed and analyzed using SPSS 26.0 and Mplus 8.3.

#### 3.3.1. Common Method Deviation Test

The Harman’s single-factor test was employed to examine common method bias. Among 51 items, a total of 7 factors with eigenvalues greater than 1 were extracted. The first factor accounted for 21.53% of the variance, which is below the critical threshold of 40%. This suggests that common method bias is not a significant concern.

#### 3.3.2. Descriptive Statistics

Marital status, service routes, domestic trade, and rank have all been converted into dummy variables. The means, standard deviations, and correlation coefficients for each variable are presented in Table 1.

#### 3.3.3. The Relationship between Seafarers’ Sense of Loneliness and Work Engagement: The Mediating Role of Executive Functions

After controlling for variables such as age, marital status, number of children, service route, and rank, a mediation effect analysis was performed. The analysis was conducted using Mplus 8.3 with bias-corrected percentile Bootstrap method for testing, with 5000 resampling iterations to calculate the 95% confidence interval. The model fit was ideal (CFI = 0.974, TLI = 0.961, RMSEA = 0.067, SRMR = 0.051). Loneliness was found to significantly negatively predict both work engagement and executive functions. Executive functions significantly positively predicted work engagement. The mediating effect of executive functions was significant (ab [indirect effect] = −0.04, SE = 0.01, 95% CI = [−0.06, −0.02]). See Figure 1,Table 2.

### 3.4. Summary

Study 1 employed relevant research methodologies to preliminarily substantiate the relationships among seafarers’ loneliness, work engagement, and executive function. However, there are three notable limitations: Firstly, Study 1 utilized self-report questionnaires to assess executive function, which reflects the level of reflective analysis in psychological evaluation. This primarily represents the extent to which individuals strive for goal achievement and rational decision making in everyday life situations. However, it does not evaluate the cognitive efficiency of individuals in specific contexts. Secondly, the findings of Study 1 were unable to differentiate the impact of the three subcomponents of executive function on work engagement. Lastly, it should be emphasized that the mediation model analysis conducted in Study 1, based on cross-sectional data, inherently limits our ability to make strong causal inferences. Cross-sectional data can only reveal correlations between loneliness, executive function, and work engagement, without establishing the direction or causality of these relationships. This limitation restricts our understanding of the underlying mechanisms. To address this, Study 2 will utilize an experimental approach to manipulate the level of loneliness among seafarers. This method will allow us to more rigorously test its causal impact on work engagement and to explore the specific mediating roles of the three subcomponents of executive function.

## 4. Study 2

### 4.1. Method

#### 4.1.1. Participants

Using G*power 3.1 software to calculate the required sample size for the experiment, with a significance level of α = 0.05 and a medium effect size (f = 0.25), it is predicted that a total sample size of 159 participants will achieve 80% statistical power. During the sailors’ land-based training, 200 sailors volunteered to participate in the experiment, including 76 captains, 65 first officers, and 59 s officers. All participants are male, aged between 35 and 45 years, and are married with children. The participants were randomly divided into two groups: the solitude group (100 participants, including 38 captains, 33 first officers, and 29 s officers) and the non-solitude group (100 participants, including 32 captains, 32 first officers, and 30 s officers).

#### 4.1.2. Experimental Design, Procedures, and Materials

A single-factor experimental design was employed, with the independent variable being the type of seafarers’ loneliness (lonely vs. non-lonely). The dependent variables included the average accuracy rate for incongruent color-word trials in the Stroop task, the average accuracy rate of correctly identified letters in the 2-back task, the switching cost in the digit transformation task, and the willingness of working engagement. The experiment is conducted on Dell computers with a resolution of 2560 × 1600 pixels and a 15.6-inch screen. It is programmed using E-Prime 3.0 software. The computer automatically records reaction times and accuracy rates. The entire experimental process takes about 30 min.

The experiment is administered individually in a laboratory setting. Participants begin by engaging in a 10 min recollection and imagination exercise based on pictures. Those in the loneliness group recall the most recent moment on board a ship when they felt most lonely, while the non-loneliness group recalls the food from the most recent lavish meal they had. Following this exercise, participants complete a self-assessment scale for loneliness. Next, they undertake three cognitive tasks: the Stroop task, the 2-back task, and the number-conversion task. To eliminate sequence effects, these tasks are presented in a random order. Finally, participants’ work engagement is measured. All participants were provided with detailed information about the study, including its purpose, procedures, potential risks, and benefits. Informed consent was obtained in accordance with standard ethical guidelines.

### 4.2. Measure

#### 4.2.1. Manipulation and Verification of Loneliness

The feeling of loneliness was induced using a recall paradigm, adapted from previous research methodologies [49]. Participants were instructed to imagine a scenario using cues based on time, location, context, and emotions. For the loneliness group, the instructions were as follows: ‘Please recall and describe in detail the most recent moment you felt lonely while working on the ship. Include the time, location, specific context, and your feelings. You have 10 min to complete this writing task.’ For the control group, the instructions were as follows: ‘Please recall and describe in detail the most recent occasion when you had a particularly lavish meal. Include the time, location, specific context, and your feelings. You have 10 min to complete this writing task.’ After completing the writing task, participants rated their current feelings of loneliness on a 7-point scale, where higher scores indicated stronger feelings of loneliness.

#### 4.2.2. Measurement of Inhibition Control

The Stroop color-word task is employed for this measure [50]. The materials consist of the Chinese characters for “red”, “yellow”, “green”, and “blue”, presented in the corresponding colors, totaling 16 items. Participants are provided with clear instructions for a color-identification task. They are to press the ‘D’ key for characters displayed in red, ‘F’ for those in yellow, ‘J’ for blue, and ‘K’ should the character appear in green. The task begins with a fixation point “+” displayed at the center of the screen for 500 milliseconds, followed by the presentation of the character. Participants must respond by pressing a key, and after the response or 2000 milliseconds, the next trial begins. Practice sessions precede the actual experiment. In the formal experiment, there are 36 congruent and 36 incongruent trials, which are presented in random order. The statistical indicator for this task is the average accuracy rate for incongruent color-word trials.

#### 4.2.3. Measurement of Working Memory

The 2-back task is used, involving 16 uppercase English letters [50]. Similar to the previous task, each trial begins with a fixation point “+” at the center of the screen for 500 milliseconds, followed by a letter. Participants are tasked with a letter comparison exercise. They must assess whether the currently presented letter is the same as the one shown two positions earlier. If they identify a match, they should press the ‘F’ key; if the letters do not match, they should press ‘J’. The next trial begins after the key response or 2000 milliseconds elapse. There is a practice session before the actual experiment, which consists of 74 judgment trials in total. The statistical indicator for this task is the average accuracy rate of correctly identified letters.

#### 4.2.4. Measurement of Cognitive Flexibility

The number-conversion task is utilized [50]. Numbers 1 to 9, excluding 5, are presented; a red number signifies judging whether the number is greater than 5 (press ‘A’) or less than 5 (press ‘L’); a blue number signifies judging whether the number is odd (press ‘A’) or even (press ‘L’). The procedure is the same as the previous tasks, starting with a 500-millisecond fixation point followed by the digit presentation. Participants must respond within 2000 milliseconds before moving to the next trial. After a practice session, the formal experiment consists of 10 mixed-task blocks, each containing 9 numbers presented randomly. The statistical measure for this task is the switch cost (the average response time for correct items in mixed tasks minus the average response time for correct items in single tasks).

#### 4.2.5. Measurement of Work Engagement

The work engagement of the subjects was measured using a situational projection paradigm adapted from Qi and her colleagues [51], designed to mitigate the social desirability effect. The material was presented in a third-person narrative, as follows: “The job is about to come to an end, and Chen Jun stands on the deck. At this moment, the sun has sunk below the sea level, leaving behind a swath of brilliant colors. The blue sea is calm, and the calls of seabirds can be heard in the distance. Looking at the scenery before him, Chen Jun begins to plan the arrangements for his next job. If you were Chen Jun, how long would you choose for your next sea outing? Please select a duration between 1 and 8 months”. The longer the chosen duration, the higher the work engagement is indicated.

### 4.3. Results and Analysis

#### 4.3.1. Validation of Loneliness Manipulation

After excluding 23 participants with an execution function accuracy below 70%, an independent samples t-test was conducted on the induced sense of loneliness for the two groups of sailors (a total of 177 individuals). The results showed that the level of loneliness induced in the solitude group (87 participants, M = 4.74, SD = 1.21) was significantly higher than that in the non-solitude group (90 participants, M = 1.97, SD = 0.89), t(175) = 17.66, *p* < 0.001, Cohen’s d = 2.62, indicating effective manipulation of sailors’ sense of loneliness.

#### 4.3.2. Correlation Analysis

SPSS 26.0 was used to conduct a Spearman correlation analysis of the seafarers’ loneliness, inhibitory control, working memory, cognitive flexibility, and work engagement. The results are presented in Table 3.

#### 4.3.3. Testing for Mediating Effects of Executive Functions

Before the analysis, all variables except loneliness were standardized. Subsequently, the multiple mediation effects of inhibitory control, working memory, and cognitive flexibility were tested using the Process 3.5 macro model 4 in SPSS 26.0. The results of the multiple mediation analysis are presented in Table 4 and Figure 2.

As shown in Table 4, the total effect of loneliness on work engagement was significant, β = −1.51, SE = 0.10, *p* < 0.001. The direct effect of loneliness on work engagement was also significant, β= −0.85, SE = 0.11, *p* < 0.001. The total indirect effect through the three mediators was significant, β = −0.65, SE = 0.10, with a 95% confidence interval (CI) ranging from −0.84 to −0.45. The indirect effects of loneliness on work engagement through each mediator were also examined. The indirect effect through inhibitory control was significant, β = −0.13, SE = 0.04, 95% CI [−0.22, −0.05], accounting for 20% of the total indirect effect. The indirect effect through working memory was significant, β = −0.15, SE = 0.05, 95% CI [−0.26, −0.05], contributing 23.1% to the total indirect effect. The indirect effect through cognitive flexibility was significant, β = −0.37, SE = 0.08, 95% CI [−0.52, −0.21], representing 56.9% of the total indirect effect. These findings suggest that cognitive flexibility plays the largest mediating role in the relationship between loneliness and work engagement.

These results suggest that loneliness has a strong total effect on work engagement, with a significant portion of this effect being mediated by inhibitory control, working memory, and cognitive flexibility. The overall model provides a robust explanation of the variance in work engagement, with a final R^2^ of 0.70.

## 5. Discussion and Conclusions

### 5.1. Predictive Role of Seafarers’ Loneliness in Work Engagement

This study forges new ground by integrating correlational surveys with behavioral experiments to substantiate the causal relationship between loneliness experienced by Chinese seafarers and their work engagement with a rich body of evidence. The findings indicate that loneliness among seafarers can significantly negatively predict their work engagement [13,52,53], corroborating research Hypothesis 1 and validating the work–psychological health model for Chinese seafarers based on the Job Demands–Resources Theory [2]. These conclusions extend the occupational applicability of this theory, illuminating that within the demographic of Chinese seafarers, loneliness similarly leads to the depletion of work engagement. The findings also reveal that seafarers who experience loneliness do not compensate for their lack of belonging by increasing work engagement. Although the social reconnection theory suggests that lonely seafarers may not invest more in relationships with colleagues, they also do not offset this by engaging more in their work. Evidently, the performance enhancement derived from work engagement is insufficient to ameliorate and heal the sense of belonging interrupted by the absence of interpersonal relationships.

The practical implication of these conclusions is that if shipping companies intend to employ performance rewards to mitigate the seafarers’ loneliness to bolster their work engagement, such measures may likely be ineffective. Additionally, while prolonged professional experience reshapes seafarers’ brain functions [23], it does not confer complete “immunity”; loneliness persistently influences seafarers’ psychological states. Shipping companies must not disregard the negative impact of loneliness on work engagement due to its objective presence in the maritime work environment. They could engage in compensating for the deficit of high-quality interpersonal relationships by exploiting online platforms or AI devices. Alleviating loneliness in this manner could enhance their work engagement and ensure navigational safety.

### 5.2. The Mediating Role of Executive Functions

Study 1 embarked on a preliminary exploration of the mediating role executive function plays between seafarers’ feelings of loneliness and their work engagement through correlational research. It principally reflects the level of seafarers’ psychological analysis and introspection, examining the extent to which individuals pursue goal achievement and make rational decisions within everyday life contexts. Study 2 advanced this understanding by experimentally manipulating seafarers’ work-related loneliness and assessing their cognitive efficiency within highly structured environments. This clarified the differential impacts of the subcomponents of executive function on the relationship between seafarers’ loneliness and work engagement. The validation of research Hypotheses 2 and Hypotheses3 provided a comprehensive measurement of behavioral tendencies and immediate optimal actions, thus enriching the inquiry into the mediating pathways of executive functions between seafarers’ sense of loneliness and their engagement at work.

#### 5.2.1. The Mediating Role of Inhibitory Control

The findings of this study highlight the significant mediating role of inhibitory control in the relationship between loneliness and work engagement among Chinese seafarers. The indirect effect of loneliness on work engagement through inhibitory control was statistically significant, accounting for 20% of the total indirect effect. This indicates that loneliness impairs seafarers’ inhibitory control abilities, which in turn reduces their level of work engagement. Inhibitory control, a key executive function, involves the ability to suppress irrelevant or distracting information and impulses to maintain focus on task-related activities [54]. The results suggest that loneliness disrupts this cognitive process, making it more difficult for seafarers to concentrate and effectively engage in their work. This disruption likely leads to decreased productivity and work performance, further exacerbating the negative effects of loneliness. These findings underscore the importance of cognitive health interventions in mitigating the impact of loneliness on work engagement. Enhancing inhibitory control through cognitive training programs or mindfulness practices could potentially buffer against the detrimental effects of loneliness, helping seafarers maintain better work engagement and overall performance. Shipping companies should consider incorporating such interventions into their mental health support strategies to foster a more resilient and effective workforce.

#### 5.2.2. The Mediating Role of Working Memory

Our study aligns with previous research on the general working population, which suggests that negative emotions can impair working memory functions [55,56]. Working memory is crucial for the temporary storage and processing of information during complex cognitive activities. It forms the cognitive foundation essential for understanding contexts, logical reasoning, and problem-solving tasks [57]. In the maritime work environment, persistent loneliness can deplete seafarers’ cognitive resources, diminishing the complex problem-solving capacity supported by working memory. Compared to the level of inhibitory control, a well-developed working memory provides seafarers with a richer pool of cognitive resources, enhancing their proactive work states. Our study specifically examined the mediating role of working memory in the relationship between loneliness and work engagement among Chinese seafarers. The findings indicate a statistically significant indirect effect of loneliness on work engagement through working memory, accounting for 22% of the total indirect effect. This underscores the critical function of working memory in maintaining work engagement, especially under ICE conditions.

#### 5.2.3. The Mediating Role of Cognitive Flexibility

Our investigation into the mediating role of cognitive flexibility revealed significant insights into its effect on the relationship between loneliness and work engagement among Chinese seafarers. The outcomes of Study 2 demonstrated that cognitive flexibility significantly mediates this relationship, explaining 25% of the total indirect effect. This finding aligns with Ahçı and his colleagues’ research [29], underscoring the importance of cognitive flexibility as an indicator of adaptability in various situations. In the maritime context, where work is often monotonous and social encounters are limited, high levels of cognitive flexibility are not typically demanded on a daily basis. However, the significance of cognitive flexibility becomes pronounced during sudden changes in routine, interpersonal conflicts, or emergencies. In such situations, the ability to adapt quickly and choose the best behavioral strategies is crucial for maintaining work engagement and ensuring safety [24,29]. Our findings indicate that loneliness impairs seafarers’ cognitive flexibility, hindering their ability to cope with unexpected challenges and conflicts, which subsequently leads to decreased work engagement. Comparatively, cognitive flexibility’s effect on work engagement was found to be higher than that of inhibitory control and working memory. While inhibitory control primarily assists in suppressing irrelevant distractions and maintaining task focus [50], and working memory supports the retention and manipulation of task-related information [55], cognitive flexibility facilitates the adaptation to novel and changing situations [25]. This adaptability is particularly crucial in the maritime environment, where unexpected changes and stressors are common. The superior effect of cognitive flexibility suggests that being able to swiftly adjust cognitive and emotional responses is more beneficial for sustaining work engagement under ICE conditions.

The implications of these findings are profound. Enhancing cognitive flexibility not only supports better emotion regulation and problem solving but also ensures that seafarers can maintain high levels of engagement and productivity even in the face of adversity. Therefore, maritime training programs should incorporate cognitive activities aimed at improving flexibility, such as mindfulness practices and adaptive thinking exercises. By doing so, shipping companies can foster a more resilient and engaged workforce, capable of handling the unique challenges of the seafaring environment.

This research represents the first laboratory-based investigation into the effect of seafarers’ loneliness on work engagement and higher cognitive functions. Our findings offer valuable insights into strategies for alleviating seafarers’ feelings of loneliness, improving their work engagement, and ensuring maritime safety. To mitigate loneliness, one effective strategy could be enhancing internet connectivity aboard ships, allowing for increased communication between sailors and their families on land. Additionally, implementing simple training programs on ships to enhance executive functions, particularly cognitive flexibility, could promote higher levels of work engagement and, thereby, enhance maritime safety. These approaches are not only cost-effective but also practical and expedient. By integrating these strategies, shipping companies can create a more resilient and engaged workforce, ultimately contributing to safer and more efficient maritime operations.

### 5.3. Limitations and Future Directions

This study employed a non-probability sampling method, selecting a sample of 1356 seafarers from five crew service companies and three shipping enterprises. While this approach ensured a sufficient sample size given the research context, it inherently carries limitations that may affect the external validity of the findings. Since the sample was not randomly selected, there is a potential for bias, particularly concerning the representativeness of the sample. Consequently, the conclusions drawn should primarily be considered applicable to populations similar to the specific group studied—seafarers with sailing experience within the past year—rather than being generalized to the broader seafarer population or other occupational groups. Specifically, the sample was chosen based on convenience and operational feasibility, which may have led to overrepresentation or underrepresentation of certain subgroups. For example, seafarers from specific functions, routes, or company sizes might not be proportionately represented in the sample, which could influence the generalizability of the results. Additionally, the voluntary nature of participation may have further exacerbated sample bias, as some seafarers may have chosen not to participate due to specific motivations or time constraints.

To mitigate the impact of these limitations, future research should consider employing probability sampling methods to enhance the representativeness of the sample. For instance, stratified random sampling could ensure that seafarers from different backgrounds and characteristics are adequately represented in the sample. Furthermore, expanding the sample source to include more regions and different types of shipping enterprises would help improve the external validity of the findings. In summary, although the non-probability sampling method used in this study presents certain limitations, the results still provide valuable insights into the specific behaviors and attitudes of the seafarer population under study, provided they are interpreted with caution. To further validate and extend these findings, subsequent research should aim to refine the sampling approach.

Additionally, due to the instability of seafarers’ work and the challenges in maintaining continuous follow-up, Study 1 employed cross-sectional data to construct a mediation model examining the relationships among loneliness, executive function, and work engagement. However, the inherent limitations of cross-sectional data preclude it from providing robust evidence for causal inferences, serving instead as a preliminary exploration of these relationships. Future research should consider utilizing longitudinal data collection methods to more thoroughly examine and establish the causal relationships within this model.

The limitation of Study 2 is that all participating mariners had over five years of professional experience. This may lead to variations in the mediating effects of the subcomponents of executive function across different career stages. Future research should therefore include early-career seafarers in the experiments to enhance the comprehensiveness of the conclusions.

### 5.4. Conclusions

The feeling of loneliness among seafarers can significantly predict a negative influence on their work engagement. The seafarers’ inhibitory control, working memory, and cognitive flexibility play a mediating role in this relationship.

## Figures and Tables

**Figure 1 behavsci-14-00880-f001:**
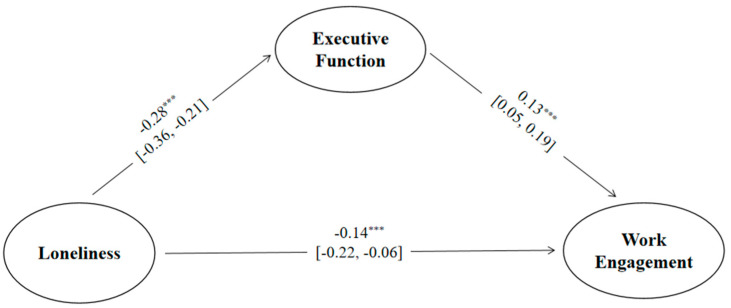
The mediating role of seafarers’ executive function between loneliness and work engagement. *** *p* < 0.001.

**Figure 2 behavsci-14-00880-f002:**
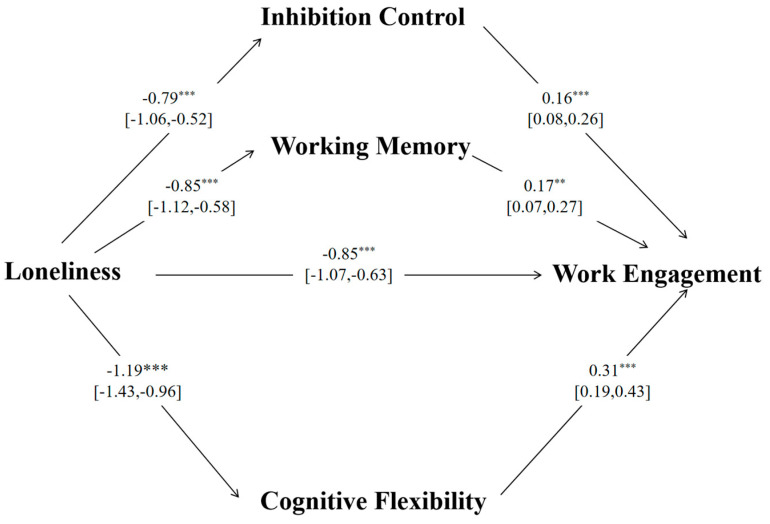
The effects of loneliness on work engagement through inhibitory control, working memory, and cognitive flexibility. ** *p* < 0.01,*** *p* < 0.001.

**Table 1 behavsci-14-00880-t001:** Descriptive statistics and correlation matrix of survey variables (n = 1231).

	*M*	*SD*	1	2	3	4	5	6	7
Loneliness	50.77	7.11	1						
Executive Function	47.01	13.46	−0.25 **	1					
Work Engagement	58.50	16.25	−0.13 **	0.13 **	1				
Age	36.67	3.77	−0.05	0.04	0.13 **	1			
Marital Status	0.63	0.46	0.03	−0.04	−0.11 **	−0.62 **	1		
Number of Children	2.76	1.31	−0.03	0.03	0.11 **	0.58 **	−0.81 **	1	
Service Routes	0.54	0.19	0.07 *	−0.09 **	0.01	−0.14 **	0.03	−0.05	
Rank	2.06	0.76	0.13 **	−0.02	−0.06 *	0.21 **	−0.22 **	0.22 *	0.03

Marital status as a dummy variable: single = 0, married = 1. Service routes as a dummy variable: domestic trade = 0, foreign trade = 1. Rank as a dummy variable: support level = 1, operational level = 2, managerial level = 3. * *p* < 0.05, ** *p* < 0.01.

**Table 2 behavsci-14-00880-t002:** Summary of standardized total, indirect, and direct effects in structural equation modeling.

Path	Estimate	S.E.	Z (Est./S.E.)	*p*-Value
Model Fit	Chi-Square			
Value	200.73			0.001
Degrees of Freedom	24			
Loneliness→Work Engagement	−0.14	0.04	−3.43	0.001
Executive Function→Work Engagement	0.13	0.04	3.57	0.001
Loneliness→Executive Function	−0.28	0.04	−6.89	0.001

**Table 3 behavsci-14-00880-t003:** Descriptive statistics and correlation matrix for variables in the experiment (n =177).

	M	SD	1	2	3	4
1. Loneliness Style	0.50	0.50				
2. Work Engagement	4.43	2.16	−0.72 **			
3. Inhibition Control	0.97	0.02	−0.42 **	0.44 **		
4. Working Memory	0.89	0.06	−0.45 **	0.49 **	0.31 **	
5. Cognitive Flexibility	0.90	0.10	−0.62 **	0.61 **	0.27 **	0.53 **

Note, loneliness style is a dummy variable: not Loneliness = 0, loneliness = 1. ** *p* < 0.01.

**Table 4 behavsci-14-00880-t004:** Summary of R^2^ values and effects.

Outcome Variable	R^2^	β	Boot *SE*	95%CI Lower	95%CI Upper	*t*
Inhibitory Control	0.16	−0.79	0.14	−1.06	−0.52	−5.71 ***
Working Memory	0.18	−0.85	0.14	−1.12	−0.58	−6.25 ***
Cognitive Flexibility	0.36	−1.19	0.12	−1.43	−0.96	−9.89 ***
Work Engagement (total)	0.57	−1.50	0.10	−1.69	−1.31	−7.49 ***
Work Engagement (final)	0.70	−0.85	0.11	−1.08	−0.63	−15.14 ***

Note, Boot SE = Bootstrap standard error; CI = confidence interval. *** *p* < 0.001.

## Data Availability

The data are available from the authors; however, they are not publicly available. Interested researchers may contact the corresponding author for access to the datasets.

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
