# Peer review of "The Effect of Loneliness on Work Engagement among Chinese Seafarers: Mediating Effects of Executive Functions"

_behavsci, 2024, doi:10.3390/bs14100880_

Round 1

Reviewer 1 Report

Comments and Suggestions for Authors

My first concerns deal with ethics.  On page 13, the authors state that they received informed consent from "face-to-face discussions" BUT in section 3.1., it is stated that the participants completed surveys through a service.  The two are impossible to co-occur. 

It is also illogical that executive function would be predicted by loneliness.  Both have a genetic component so the two should be treated as individual predictors of work engagement and the model in Figure 1 should be scrapped.  It is also very questionable to conduct a mediation analysis with cross-sectional data.  

Study 2 may be what the authors were referring to in their ethics statement, but it is strange that verbal assent was obtained as they were in-person and written consent would have been preferable.  More detail is needed to explain how loneliness was manipulated in section 4.2.1. Feeling "alone" as state in line 262 is not the same as feeling lonely.  An individual may want to be alone to enjoy solitude and therefore is not lonely.

Comments on the Quality of English Language

Other than some spelling issues (e.g., see line 252), the paper is fine.

Author Response

Response to Reviewer 1 Comments

1. Summary

Thank you very much for taking the time to review this manuscript. Your valuable feedback has greatly inspired us, helping us make further progress in academic paper writing. We have made revisions and additions based on the valuable feedback you provided. We believe that these revisions will greatly contribute to the improvement of the paper’s quality.

2. Point-by-point response to Comments and Suggestions for Authors

Comments 1: My first concerns deal with ethics.  On page 13, the authors state that they received informed consent from "face-to-face discussions" BUT in section 3.1., it is stated that the participants completed surveys through a service.  The two are impossible to co-occur.

Response 1:

Thank you for diligently pointing out the issues in our manuscript. Your feedback is invaluable for enhancing the rigor of our research. We acknowledge that there was an oversight in our initial submission where we did not separately explain the procedures for obtaining informed consent in Study 1 and Study 2. In Study 1, we first obtained verbal consent from the shipping company online, followed by the collection of written informed consent from the crew members after they completed their work voyage. In Study 2, informed consent was obtained directly through face-to-face interactions. To further improve the transparency and rigor of our study, we have included scanned copies of the original informed consent forms in the non-published material for your review.(informed consent form (study1,2))

We have also made corresponding additions and revisions in the manuscript at lines 154-156 and lines 542-547.

Comments 2: It is also illogical that executive function would be predicted by loneliness. Both have a genetic component so the two should be treated as individual predictors of work engagement and the model in Figure 1 should be scrapped. It is also very questionable to conduct a mediation analysis with cross-sectional data.  

Response 2: 

Thank you very much for taking the time out of your busy schedule to review our paper. Your feedback provides important guidance for further improving our research.

1. Regarding the Relationship Between Executive Function and Loneliness

We understand your concerns regarding the logical premise of loneliness predicting executive function. Our hypothesis that “loneliness may impair executive function” is based on previous research conducted on general populations. For instance, studies by Ahçı, S. A. Z. G. (2023), Dickinson et al. (2011), Sims et al. (2011), and Evans et al. (2019) have found that loneliness can reduce individuals’ inhibitory control, working memory, and cognitive flexibility.

Theoretically, we rely on the “Limited Cognitive Resource Theory,” which posits that individuals, in coping with the emotional distress caused by loneliness, may allocate their limited cognitive resources to emotional regulation. Therefore, we hypothesize that “loneliness could impair executive function.”

In our hypothesis that “reduced executive function lowers work engagement,” we primarily draw on the “Job Demands-Resources Theory.” Executive functions can be conceptualized as a set of personal resources that facilitate coping with occupational demands, thereby potentially enhancing seafarers’ work engagement. Additionally, research by Tan et al. (2022) on general employees found a negative relationship between executive function and work engagement. Based on these studies, we propose that executive function mediates the relationship between seafarers’ loneliness and work engagement.

2. Regarding Mediation Analysis:

You raised an important question about the appropriateness of conducting mediation analysis using cross-sectional data. This is indeed a critical issue. The reasons for using cross-sectional data in Study 1 to conduct mediation analysis are as follows:

Firstly, in Study 1, we employed cross-sectional data to preliminarily explore the potential relationships between loneliness, executive function, and work engagement. The goal was not to establish strong causal inferences but rather to identify possible associations. Based on the findings from Study 1, we proceeded with Study 2, which involved a behavioral experiment. This sequential approach helps to avoid the higher costs associated with conducting an experiment right from the outset. Furthermore, the classic steps for mediation analysis proposed by Baron and Kenny (1986) were originally developed within a framework of cross-sectional research. Many other studies also use cross-sectional data for mediation analysis as a method to preliminarily determine relationships between variables (e.g., Kim et al., 2019; King, 2016; Miconi et al., 2020).

However, we acknowledge that the results from cross-sectional data cannot provide strong evidence for causal inferences. Therefore, in the conclusion section of Study 1, we have included a discussion of the limitations of this approach and emphasized the necessity of Study 2 (the behavioral experiment).

The specific changes are as follows:

Lastly, and most importantly, the mediation model analysis in Study 1, based on cross-sectional data, can only indicate the correlation between loneliness, executive function, and work engagement, and cannot provide strong evidence for causal inference. Therefore, Study 2 will employ experimental manipulation of seafarers’ loneliness to further investigate its impact on work engagement and explore the specific mediating roles of the three subcomponents of executive function. (page6,line225-231)  

References:

Ahçı, S. A. Z. G. (2023). The role of cognitive flexibility and hope in the relationship between loneliness and psychological adjustment: a moderated mediation model. Educational and Developmental Psychologist,40(1),74–85.

Douglas H. Clements, J. S. C. G. (2016). Learning executive function and early mathematics: Directions of causal relations. Early Childhood Research Quarterly,36, 79-90.

Sims, R. C., Levy, S. A., Mwendwa, D. T., Callender, C. O., & Campbell, A. J. (2011). The influence of functional social support on executive functioning in middle-aged African Americans . Aging Neuropsychology and Cognition, 18(4), 414-431. http://doi.org/10.1080/13825585.2011.567325

Dickinson, W. J., Potter, G. G., Hybels, C. F., McQuoid, D. R., & Steffens, D. C. (2011). Change in stress and social support as predictors of cognitive decline in older adults with and without depression. International Journal of Geriatric Psychiatry, 26(12), 1267-1274. http://doi.org/10.1002/gps.2676

Evans, I. E. M., Martyr, A., Collins, R., Brayne, C., & Clare, L. (2019). Social Isolation and Cognitive Function in Later Life: A Systematic Review and Meta-Analysis. Journal of Alzheimer's Disease, 70(s1), S119-S144. http://doi.org/10.3233/JAD-180501

Tan, C., Nasir, H., Pheh, K., Cong, C. W., Tay, K., & Cheong, J. (2022). The Mediating Role of Work Engagement in the Relationship between Executive Functioning Deficits and Employee Well-Being. International Journal of Environmental Research and Public Health, 19(20), 13386. http://doi.org/10.3390/ijerph192013386

Baron, R. M., & Kenny, D. A. (1986). The moderator-mediator variable distinction in social psychological research: Conceptual, strategic, and statistical considerations. Journal of Personality and Social Psychology, 51(6), 1173-1182. https://doi.org/10.1037/0022-3514.51.6.1173

Comments 3: Study 2 may be what the authors were referring to in their ethics statement, but it is strange that verbal assent was obtained as they were in-person and written consent would have been preferable.  More detail is needed to explain how loneliness was manipulated in section 4.2.1.

Response 3: 

Thank you very much for your valuable feedback, which has greatly contributed to improving the rigor and reproducibility of our research. Based on your suggestions, we have uploaded the original scanned copies of the informed consent forms from Study 1 and Study 2 as attachments for your review.

Additionally, in section 4.2.1, we have provided a more detailed explanation of the manipulation of loneliness, and we have uploaded the experimental materials for your reference. The specific revisions are as follows:

4.2.1 Manipulation and Verification of Loneliness.   

The feeling of loneliness was induced using a recall paradigm, adapted from previous research methodologies[49]. Participants were instructed to imagine a scenario using cues based on time, location, context, and emotions. For the loneliness group, the instructions were as follows: “Please recall and describe in detail the most recent moment you felt lonely while working on the ship. Include the time, location, specific context, and your feelings. You have 10 minutes to complete this writing task.” For the control group, the instructions were: “Please recall and describe in detail the most recent occasion when you had a particularly lavish meal. Include the time, location, specific context, and your feelings.You have 10 minutes to complete this writing task.” After completing the writing task, participants rated their current feelings of loneliness on a 7-point scale, where higher scores indicated stronger feelings of loneliness.(line264-275)

Comments 4: Feeling "alone" as state in line 262 is not the same as feeling lonely.  An individual may want to be alone to enjoy solitude and therefore is not lonely.

Response 4:

Thank you very much for taking the time to review our paper amidst your busy schedule, and we are grateful for your help in identifying the error. This was an oversight in our writing process, and we have now corrected ‘alone’ to ‘lonely’.

4. Response to Comments on the Quality of English Language

Point 1: Comments on the Quality of English Language

Other than some spelling issues (e.g., see line 252), the paper is fine.

Response 1:  It has been revised in the manuscript.

Reviewer 2 Report

Comments and Suggestions for Authors

Dear Author, you have done a nice job. I have a few suggestions and added them in the word document.

Best regards,

Author Response

Response to Reviewer 2 Comments

1. Summary

Thank you very much for taking the time to review this manuscript. Your feedback has been the greatest encouragement and help to us. We have made detailed revisions based on your suggestions. Please find the detailed responses below and the corrections highlighted in the re-submitted files.

2. Point-by-point response to Comments and Suggestions for Authors

Comments 1:  The introduction is quite lengthy and dense, which may make it difficult for readers to follow the key points. It could be more concise. Some sentences are complex and could be simplified for better readability (Paragraph 1, line 22-24 “ Safety is…”, line 25-27 “Work engagement…”, Paragraph 2, line 33-35, Paragraph 3, line 52-54, Paragraph 4, line 74-78, Paragraph 5 line 85-87, Paragraph 6 line 118-121, Paragraph 7 Line 145-148).

Response 1:

Thank you very much for your valuable feedback, which has been extremely helpful in improving the quality of our paper. We have revised all the sentences you pointed out, ensuring brevity while also enhancing readability. The revised sentences have been highlighted in the original manuscript. (They are located respectively at original Paragraph 1, line 22-24 “ Safety is…”, line 25-27 “Work engagement…”, Paragraph 2, line 33-35, Paragraph 3, line 52-54, Paragraph 4, line 74-78, Paragraph 5 line 85-87, Paragraph 6 line 118-121, Paragraph 7 Line 145-148).

Comments 2: Also, there is some repetition of ideas, particularly regarding the impact of loneliness on work engagement and the role of executive functions. My suggestions for improvement are: reduce the length by focusing on the most critical points and eliminating redundant information and use simpler sentences and clearer language to improve readability.

Response 2: 

Thank you very much for your valuable feedback. We have made detailed revisions to the theoretical framework section, removing some repetitive statements and redundant content, striving for accuracy and logical clarity in our language. The detailed revisions can be found highlighted in the re-submitted files. For example, we have deleted the original line 53, reviseline384-389,deleted the original line 443-444,deleted the original line458-459

3. Response to Comments on the Quality of English Language

Point 1: Comments on the Quality of English Language

Other than some spelling issues (e.g., see line 252), the paper is fine.

Response 1:  The spelling issue in line 252 has been corrected to ‘the willingness of working engagement’

Reviewer 3 Report

Comments and Suggestions for Authors

In your study, you mention as experimental design procedure 'convenience sampling'. That is, you get what you can get. There is no, as a statistical method, such thing as 'convenience sampling', even if you can see in some published papers. If you state that you got data in such way, in theory you can not use inferential methods; your study could be applied to your data set, without the possibility of generalization. In the abstract, avoid using the word 'convenience', and in the text in 3.1. Also, when you use 'correlation' surely you mean relation. Correlation is a statistical term; Pearson's correlations are a measure of inter-relations (not causal relations) between numerical variables; when these are ordinal, some others measures such as Spearman correlations, are preferable.

In table 1, surely Rank means Job Level; but his is a non-numerical variable, so care should be taken when employing numerical-variables techniques.

In figure 1 you present a SEM with three latent variables, with a negative influence of loneliness. If you use SPSS Amos in estimating the model, you should get a chi-square statistics for overall fit (non reported) and Z type statistics for tests over the structural coefficients (instead you report confidence interval for these, while in table 5 you include these statistics and their corresponding p-values). Also, tests for the influence between latent and observable variables, with the direction of causality.

In figure 2 you present the second SEM, but in this case, where are the goodness of fit measures?

In part 5, 'Discussion', your present the results and their conclusions. Why don't you name point 5 as 'Discussion and conclusions', so you can omit a very poor and short point 6 'Conclusions'; these lines could be integrated in the previous point as and ending to it.

Author Response

Response to Reviewer 3 Comments

1. Summary

Thank you very much for taking the time to review this manuscript. Your valuable feedback has greatly inspired us, helping us make further progress in academic paper writing. We have made revisions and additions based on the valuable feedback you provided.We believe that these revisions will greatly contribute to the improvement of the paper’s quality.

2. Point-by-point response to Comments and Suggestions for Authors

Comments 1: In your study, you mention as experimental design procedure 'convenience sampling'. That is, you get what you can get. There is no, as a statistical method, such thing as 'convenience sampling', even if you can see in some published papers. If you state that you got data in such way, in theory you can not use inferential methods; your study could be applied to your data set, without the possibility of generalization. In the abstract, avoid using the word 'convenience', and in the text in 3.1.

Response 1

Thank you very much for your valuable feedback and insightful comments regarding the use of “convenience sampling” in our study. We appreciate your clarification on the limitations associated with this sampling method, particularly concerning its impact on the generalizability of our findings.

In response to your comments, we will revise the manuscript to address these concerns. Specifically, we will avoid the term “convenience sampling” in the abstract. We will remove the term from the abstract and replace it with a more appropriate description of our sampling method, ensuring that it aligns with the methodological rigor expected in inferential studies.

We will clarify the limitations in Section 3.1: In Section 3.1, we will provide a more detailed explanation of our sampling strategy and explicitly acknowledge the limitations this method imposes on the generalizability of our results.We will also discuss how this may affect the interpretation of our findings and their applicability beyond our specific sample. Additionally, the limitations of drawing conclusions from this sampling method are discussed in section 5.3.

5.3 Limitations and Future Directions

This study employed a non-probability sampling method, selecting a sample of 1,356 seafarers from five crew service companies and three shipping enterprises. While this approach ensured a sufficient sample size given the research context, it inherently carries limitations that may affect the external validity of the findings. Since the sample was not randomly selected, there is a potential for bias, particularly concerning the representativeness of the sample. Consequently, the conclusions drawn should primarily be considered applicable to populations similar to the specific group studied—seafarers with sailing experience within the past year—rather than being generalized to the broader seafarer population or other occupational groups. Specifically, the sample was chosen based on convenience and operational feasibility, which may have led to overrepresentation or underrepresentation of certain subgroups. For example, seafarers from specific functions, routes, or company sizes might not be proportionately represented in the sample, which could influence the generalizability of the results. Additionally, the voluntary nature of participation may have further exacerbated sample bias, as some seafarers may have chosen not to participate due to specific motivations or time constraints.

To mitigate the impact of these limitations, future research should consider employing probability sampling methods to enhance the representativeness of the sample. For instance, stratified random sampling could ensure that seafarers from different backgrounds and characteristics are adequately represented in the sample. Furthermore, expanding the sample source to include more regions and different types of shipping enterprises would help improve the external validity of the findings.

In summary, although the non-probability sampling method used in this study presents certain limitations, the results still provide valuable insights into the specific behaviors and attitudes of the seafarer population under study, provided they are interpreted with caution. To further validate and extend these findings, subsequent research should aim to refine the sampling approach.

Comments 2: Also, when you use 'correlation' surely you mean relation. Correlation is a statistical term; Pearson's correlations are a measure of inter-relations (not causal relations) between numerical variables; when these are ordinal, some others measures such as Spearman correlations, are preferable.

Response 2: 

Thank you very much for your valuable feedback. We have revised the relevant analysis method to Spearman correlation. The specific content is in section 4.3.2 Correlation Analysis, and the results are presented in Table 2.

Table 2. Descriptive Statistics and Correlation Matrix for Variables in the Experiment(n =177)

M

SD

1

2

3

4

1.Loneliness Style

0.50

0.50

2.Work Engagement

4.43

2.16

-0.72**

3.Inhibition Control

0.97

0.02

-0.42**

0.44**

4.Working Memory

0.89

0.06

-0.45**

0.49**

0.31**

5.Cognitive Flexibility

0.90

0.10

-0.62**

0.61**

0.27**

0.53**

*Note,Loneliness style as a dummy variable: Not Loneliness = 0, Loneliness = 1

Comments 3: In table 1, surely Rank means Job Level; but his is a non-numerical variable, so care should be taken when employing numerical-variables techniques.

Response 3:

 Thank you for your review of our research and your valuable suggestions. We have noted the issue you raised regarding the Rank variable being non-numerical. To ensure the accuracy of our analysis, we have encoded the Rank variable as dummy variables, making it suitable for numerical-variable techniques. We have updated the model accordingly and clearly explained this process in the text. We hope this revision addresses the concerns you raised.The explanation is provided in lines 183-184 of the revised manuscript.

Comments 4: In figure 1 you present a SEM with three latent variables, with a negative influence of loneliness. If you use SPSS Amos in estimating the model, you should get a chi-square statistics for overall fit (non reported) and Z type statistics for tests over the structural coefficients (instead you report confidence interval for these, while in table 5 you include these statistics and their corresponding p-values). Also, tests for the influence between latent and observable variables, with the direction of causality.

Response 4: 

Thank you for reviewing our research and providing valuable suggestions. We have included the necessary details regarding the structural equation model (SEM) and have reorganized this data into a more readable format (see new Table 2) in the revised manuscript.

Chi-Square Test of Model Fit:

Value: 200.728

Degrees of Freedom: 24

P-Value: 0.0000

The Chi-Square statistic indicates that the model does not perfectly fit the data, as the P-value is less than 0.05. However, this should be interpreted with caution due to the test’s sensitivity to large sample sizes.

Z-Statistics and P-Values for Structural Coefficients:

We have included the estimates, standard errors (S.E.), Z-statistics (Est./S.E.), and corresponding P-values for key paths in the model:

IS → EN: Estimate = -0.143, Z = -3.430, P = 0.001

EXCU → EN: Estimate = 0.128, Z = 3.573, P = 0.000

IS → EXCU: Estimate = -0.282, Z = -6.895, P = 0.000

Causal Direction Tests Between Latent and Observed Variables:

The total effect of IS on EN is negative (Estimate = -0.179, Z = -4.459, P = 0.000).

Specific indirect effect through EXCU (Estimate = -0.036, Z = -3.249, P = 0.001).

Direct effect of IS on EN (Estimate = -0.143, Z = -3.430, P = 0.001).

Table 2. Summary of Standardized Total, Indirect, and Direct Effects in Structural Equation Modeling

Path

Estimate

S.E.

Z (Est./S.E.)

P-Value

Model Fit

Chi-Square

Value

200.73

0.001

Degrees of Freedom

24

Loneliness→Work Engagement

-0.14

0.042

-3.43

0.001

Executive Function→Work Engagement

0.13

0.036

3.57

0.001

Loneliness→Executive Function

-0.28

0.041

-6.89

0.001

Comments 5: In figure 2 you present the second SEM, but in this case, where are the goodness of fit measures?

Response 5: 

Thank you very much for your valuable feedback. We have revised the original Table 3 by adding the R² metric as part of the fit measures, creating a new Table 4. The specific changes can be found in the revised manuscript on lines 364-369.

Table 4. Summary of R² Values and Effects

Outcome Variable

R2

β

Boot SE

95%CI Lower

95%CI Upper

t

Inhibitory Control

0.16

-0.79

0.14

-1.06

-0.52

-5.71***

Working Memory

0.18

-0.85

0.14

-1.12

-0.58

-6.25***

Cognitive Flexibility

0.36

-1.19

0.12

-1.43

-0.96

-9.89***

Work Engagement(total)

0.57

-1.50

0.10

-1.69

-1.31

-7.49***

Work Engagement(final)

0.70

-0.85

0.11

-1.08

-0.63

-15.14***

*Note. Boot SE= Bootstrap Standard Error; CI= Confidence Interval

Comments 6: In part 5, 'Discussion', your present the results and their conclusions. Why don't you name point 5 as 'Discussion and conclusions', so you can omit a very poor and short point 6 'Conclusions'; these lines could be integrated in the previous point as and ending to it. 

Response 6: 

Thank you very much for your valuable feedback. We have renamed Part 5 to ‘5. Discussion and Conclusions’,deleted Part 6, and added Sections ‘5.3 Limitations and Future Directions’ and ‘5.4 Conclusions’.

Round 2

Reviewer 1 Report

Comments and Suggestions for Authors

Unfortunately you have not addressed my concerns and keep the mediation analyses with cross-sectional data and add confusion regarding informed consent.

Comments on the Quality of English Language

Fine

Author Response

  1. Regarding the informed consent

Thank you for your thorough and insightful comments regarding the ethical considerations in our research. We appreciate the opportunity to clarify and enhance the clarity of our manuscript.We understand the concern regarding the potential inconsistency in our description of the informed consent process. To clarify:

For Study 1: Because seafarers work in offshore or international waters, we first obtained verbal consent from the shipping company online. After the crew members completed their voyage and returned to land, we then collected their written informed consent. ï¼ˆsee line155-158 )

For Study 2: In Study2, the participants were provided with detailed information about the study, including its purpose, procedures, potential risks, and benefits. Informed consent was obtained in accordance with standard ethical guidelines. (see line271-273,565-571)

We have now revised the manuscript to clearly delineate the procedures for each study. Additionally, as previously mentioned, we have included scanned copies of the informed consent forms in the supplementary materials to provide further transparency. We hope these clarifications address your concerns, and we are open to any further suggestions you may have.

2.Regarding the Use of Cross-Sectional Data for Mediation Analysis:

We fully understand and appreciate your concerns regarding the limitations of using cross-sectional data for mediation analysis, particularly the challenge of inferring causality. We would very much like to follow your suggestion to use longitudinal data instead of cross-sectional data. However, due to practical constraints, it has been over a year and a half since the last data collection. Many of the seafarers who participated in the initial survey have changed shipping companies or are currently on leave, making it difficult to contact them and collect additional data in a short period. 

The cross-sectional data used in Study 1 was primarily an exploratory step to identify potential associations between loneliness, executive function, and work engagement. This was intended to be the first step in exploring these relationships. Many previous studies(Ginevra et al., 2016; Kim et al., 2019; Miconi et al., 2020)have similarly used cross-sectional data to investigate mediation relationships as an initial step in exploring the relationships between variables. Maxwell et al. (2011) also mentioned that cross-sectional data can still be used for exploratory analysis, especially when resources or time are limited.

We hope to conduct more comprehensive longitudinal studies in the future to further investigate these findings. We are committed to taking your valuable advice into account and will endeavor to collect longitudinal data to strengthen the causal inferences in future research.

While we recognize that the current mediation model based on cross-sectional data has its limitations, Therefore, we have thoroughly discussed the limitations of the model constructed using cross-sectional data in Study 1 within the limitations and discussion section, and highlighted the necessity of using longitudinal data in future research. (see lines 230-239,548-555).

LINE 230-239: Lastly, it should be emphasized that the mediation model analysis conducted in Study 1, based on cross-sectional data, inherently limits our ability to make strong causal inferences. Cross-sectional data can only reveal correlations between loneliness, executive function, and work engagement, without establishing the direction or causality of these relationships. This limitation restricts our understanding of the underlying mechanisms.To address this, Study 2 will utilize an experimental approach to manipulate the level of loneliness among seafarers. This method will allow us to more rigorously test its causal impact on work engagement and to explore the specific mediating roles of the three subcomponents of executive function.

LINE548-555:   Additionally, due to the instability of seafarers’ work and the challenges in maintaining continuous follow-up, Study 1 employed cross-sectional data to construct a mediation model examining the relationships among loneliness, executive function, and work engagement. However, the inherent limitations of cross-sectional data preclude it from providing robust evidence for causal inferences, serving instead as a preliminary exploration of these relationships. Future research should consider utilizing longitudinal data collection methods to more thoroughly examine and establish the causal relationships within this model.

We hope that this revised approach, along with our commitment to improving the study design in future research, addresses your concerns. We are grateful for your guidance and look forward to any further feedback you may have.

References:

Elizabeth A Stuart, Ian Schmid, Trang Nguyen, Elizabeth Sarker, Adam Pittman, Kelly Benke, Kara Rudolph, Elena Badillo-Goicoechea, Jeannie-Marie Leoutsakos, Assumptions Not Often Assessed or Satisfied in Published Mediation Analyses in Psychology and Psychiatry, Epidemiologic Reviews, Volume 43, Issue 1, 2021, Pages 48–52, https://doi.org/10.1093/epirev/mxab007

Ginevra, M. C., Pallini, S., Vecchio, G. M., Nota, L., & Soresi, S. (2016). Future orientation and attitudes mediate career adaptability and decidedness. Journal of Vocational Behavior, 95-96, 102-110. http://doi.org/https://doi.org/10.1016/j.jvb.2016.08.003

Kim, D. H., Bassett, S. M., So, S., & Voisin, D. R. (2019). Family stress and youth mental health problems: Self-efficacy and future  orientation mediation. American Journal of Orthopsychiatry, 89(2), 125-133. http://doi.org/10.1037/ort0000371

Miconi, D., Oulhote, Y., Hassan, G., & Rousseau, C. (2020). Sympathy for violent radicalization among college students in Quebec (Canada): The protective role of a positive future orientation. Psychology of Violence, 10(3), 344-354. http://doi.org/10.1037/vio0000278
